# Research on Double-Stage and Multi-Stage Capacitive Deionization Absorption Air-Conditioning System

Feng Cheng [†][ID], Boqing Ding [†] and Xiuwei Li *,[†]

College of Energy and Power Engineering, Nanjing University of Science and Technology, Nanjing 210094, China; fengcheng@njust.edu.cn (F.C.); boqingding123@163.com (B.D.)
* Correspondence: good3000best@163.com; Tel.: +86-25-83792722
† These authors contributed equally to this work.

**Abstract:** An absorption air-conditioning system is a good choice for green buildings. It has the superiority in the utilization of renewable energy and the refrigerant is environment-friendly. However, the performance of the traditional absorption system has been restricted by the energy waste in the thermal regeneration process. Capacitive deionization (CDI) regeneration is proposed as a potential method to improve system efficiency. In the new method-based air-conditioning system, strong absorbent solutions and pure water are acquired with the joint work of two CDI units. Nevertheless, the practical CDI device is composed of a lot of CDI units, which is quite different from the theoretical model. To reveal the performance of multiple CDI units, the model of the double/multi-stage CDI system has been developed. Analysis has been made to expose the influence of some key parameters. The results show the double-stage system has better performance than the single-stage system under certain conditions. The coefficient of performance (COP) could exceed 4.5, which is higher than the traditional thermal energy-driven system, or even as competitive as the vapor compression system. More stages with proper voltage distribution better the performance. It also provides the optimization method for the multi-stage CDI system.

**Keywords:** energy conservation; absorption air-conditioning; regeneration; capacitive deionization; double-stage systems; multi-stage systems





## 1. Introduction

At present, the combustion of fossil fuels brings huge pressure to the energy and environmental system. Energy conservation is an effective way to meet this challenge and it has become a hot topic all around the world. Buildings are among the biggest energy consumers, making them more energy-saving is of great concern [1,2]. In buildings, air-conditioning systems take up more than 40% of the energy [3], so it would be meaningful to make those systems more energy conservative. The most widely used system is the vapor compression air-conditioning system, which heavily depends on electric power and the refrigerant causes environmental problems. As an alternative, the absorption air-conditioning system could be a better choice as it favors renewable energy and the refrigerant is more environment-friendly [4,5]. However, low efficiency restricts the development of an absorption air-conditioning system. The major cause is the energy waste in the traditional thermal regeneration method [5]. For the purpose of improvement, many research studies have been carried out. Han et al. [6] proposed an absorption–compression refrigeration system based on cascade use of mid-temperature waste heat. With the same waste heat input, the proposed system generates 46.7% more cooling energy than the conventional ammonia-water absorption refrigerator. Jain et al. [7] made a comparison between the vapor compression–absorption cascaded refrigeration system and conventional vapor compression system. The results show that the electric power consumption of the cascaded system is reduced by 61% and the coefficient of performance (COP) is improved by 155%. Avanessian, Li, Bouaziz et al. [8–10] present works on single and double effect LiBr-$H_2O$

absorption system. Membrane regeneration is another choice, Li et al. [11,12] proposed a membrane regeneration absorption system based on electrodialysis (ED) technology. The theoretical COP could reach 6 under certain working conditions. Nevertheless, the experimental results show that the actual energy efficiency is much lower than the theoretical value [13]. It is mainly caused by the energy loss in membrane resistance heating and electrochemical reactions under high voltages. The membrane regeneration system avoids the energy waste in heating mode, which has a certain referential significance for developing other electric adsorption technology used for the absorption system. An example is the absorption system based on the capacitive deionization (CDI) regeneration method [14,15]. CDI is an emerging technology of desalination [16,17]. Under the electric field, ions will move to and be adsorbed by electrodes, and thus lowers the concentration and produces pure water. In the reverse process without the electric power, the ions adsorbed in the electrodes will be easily washed out by the passing solution, which enhances the concentration and realizes regeneration. The CDI regeneration method proceeds with the joint work of two CDI units [14,15]. They replace the generator and condenser in the traditional absorption air-conditioning system, the absorption and evaporation process are the same. Compared with the ED method, the CDI method has a better performance by replacing the membranes with electrodes. It eliminates the energy waste and mass transfer resistance caused by membranes. It saves the energy of electric chemical reactions. This is because the potential difference would not exceed the Nernst voltage or the standard electrode potential of water, so no hydrolysis will happen [18]. Energy recovery is another advantage of this method: the energy will be stored in the capacitor (electrode) when the CDI units are charged, by connecting the two units through an external circuit, the stored energy can be reused [19,20]. The analysis exhibits a good prospect, the COP can exceed 6 due to recovered energy [14,15]. With the temperature decreasing of the produced chilled water, the cooling load will increase. This means more water vapor (refrigerant) would be absorbed by the absorbent solution, which further reduces the concentration of the solution to be concentrated. This will improve the efficiency as the COP is higher with a lower concentration of the regenerated solution. For the same reason, with the increasing part-load ratio, COP also gets higher. This indicates that the system has favorable characteristics under high load conditions.

The present work only focuses on the situation the system only has one CDI unit for the regeneration or deionization process, while the practical CDI device is probably composed of many units or units of multi-stages [21–23]. When many CDI units work together, which parameter is the most influential one? How can we optimize the system performance? Those problems need answers. The research about what effect multiple units will have on the system is a valuable issue. Therefore, we have first investigated the double-stage system as a preliminary model. On this basis, the multi-stage system is further analyzed. Models have been developed and comparison analysis has been conducted. The results show double-stage system performs better with optimized parameters, which indicates the increase of the number of stages is helpful for performance improvement.

## 2. Materials and Methods

### 2.1. Principle of the Capacitive Deionization Process

The process of CDI water treatment can be divided into two steps as shown in Figure 1 [24,25]. When the electrodes are charged, the ions are adsorbed in the electrochemical double layer (EDL) on the surface of electrodes and produce deionized water at the outlet. When the CDI unit works in the regeneration step, the electrodes are depolarized and a wash solution is circulated. Ions are desorbed and enter into the bulk of the solution, resulting in a stream of higher concentration.

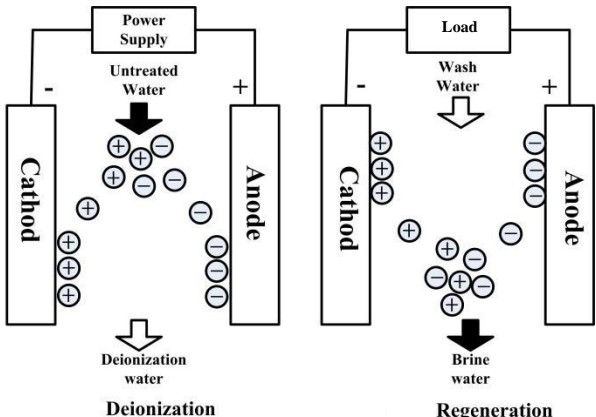

**Figure 1.** Principle of the capacitive deionization process.

The theory of EDL is the base of the CDI adsorption process: when the electrodes are charging, the surface of electrodes will attract the ions of opposite polarity to the phases interface, forming an ion area with the opposite charge, and the whole system is electrically neutral. Take the symmetrical electrolyte as an example. Assume $c(x)$ is the ion density at $x$ direction from the electrode surface, and it is the function of $\varphi(x)$, the potential at $x$ direction from the electrode surface and $T$, the temperature of the solution [22,23]:

$$c(x) = c_0 \exp\left(-\frac{ze\varphi(x)}{kT}\right). \tag{1}$$

The charge density $\rho(x)$ at $x$ direction from the surface of the electrode is:

$$\rho(x) = zec_0 \exp\left(-\frac{ze\varphi(x)}{kT}\right) \tag{2}$$

where $z$ is the valence, $e$ is the electronic charge. According to the principle of electrostatics, [24,25] the charge density and potential follow a Poisson distribution:

$$\frac{\mathrm{d}^2\varphi(x)}{\mathrm{d}x^2} = -\frac{\rho(x)}{\varepsilon} = -\frac{1}{\varepsilon}zec_0 \exp\left(-\frac{ze\varphi(x)}{kT}\right) \tag{3}$$

where $\varepsilon$ is the permittivity of a capacitor. The boundary conditions are:

$$\begin{cases} \varphi(x) = \varphi_0; x = 0 \\ \varphi(x) = 0; x = \infty \end{cases} \tag{4}$$

where $c_{ad}$, the number of adsorbed ions per square meters in EDL, is:

$$c_{ad} = -\int_0^\infty c(x)dx = -\int_0^\infty c_0 \exp\left(-\frac{ze\varphi(x)}{kT}\right)\mathrm{d}x \tag{5}$$

where $c_0$ is the initial molar concentration, mol/L. Combined Equations (3)–(5):

$$c_{ad} = \frac{(8\varepsilon kTc_0)^{1/2}}{ze}\sinh\left(\frac{ze\varphi_0}{2kT}\right) \tag{6}$$

The molar mass of adsorbed ions per square meters is:

$$m_{ad} = \frac{(8\varepsilon kTc_0)^{1/2}}{zeN_A}\sinh\left(\frac{ze\varphi_0}{2kT}\right) \tag{7}$$

where $N_A$ is the Avogadro constant. With Equation (7), the molar mass of adsorbed ions can be calculated and it is useful to analyze the deionization and regeneration processes.

### 2.2. Single-Stage CDI Absorption Air-Conditioning System

The flow chart is shown in Figure 2. The absorption and evaporation processes are the same as the traditional system. The regeneration is completed with the joint work of two CDI units. The two units alternate their roles as the regenerator and deionizer to produce both the strong absorbent solution and pure water. For example, when CDI unit 1 works as the regenerator, its electrodes are saturated with ions at the beginning. The weak solution from the absorber is sent to CDI Unit 1, where the electrodes release the ions and regenerate the solution. Then the strong solution is sent back to the absorber (Valves 1 and 5 are open, Valve 4 is closed). In the thermal energy-driven method, the solution is regenerated by getting off the absorbed water. The solution concentration and amount are equal to that of the strong solution before absorption. However, in the CDI regeneration process, the absorbed water is retained and more solutes are added from the CDI unit. When the solution becomes strong, the total solution amount actually increases with the absorbed water and added solutes. In other words, an extra amount of strong solution is acquired every turn. The extra part will be stored in Solution Storage Tank 1 (Valve 2 is open, Valve 3 is closed). During this period, CDI Unit 1 also discharges the stored energy to CDI Unit 2 with a DC/DC converter. At the same time, CDI Unit 2 works as the deionizer. Solution Storage Tank 2 is full of strong solution at the beginning. The solution from Solution Storage Tank 2 cycles between the deionizer and Solution Storage Tank 2 (Valves 7, 8 and 9 are open, Valve 9, 10 are closed). After many turns, all the solutions in Solution Storage Tank 2 become purified water and the electrodes of CDI unit 2 are saturated with ions, while CDI unit 1 has released all the absorbed ions and could not regenerate anymore. At this time, open Valve 9 and the water is sent to Water Storage Pool for utilization. Then CDI unit 1 and 2 will alternate their roles and the cycle continues.

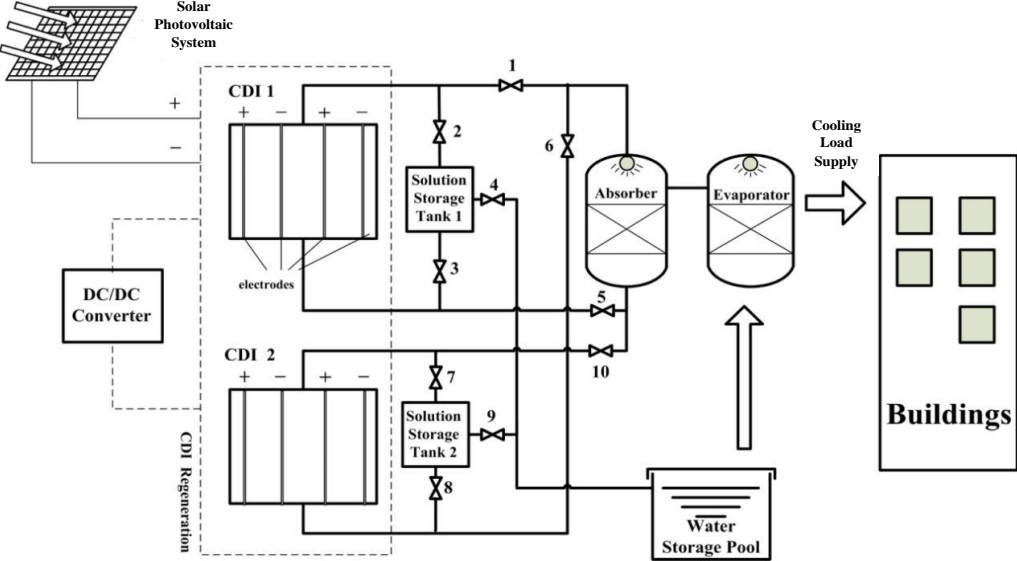

**Figure 2.** Flow chart of the single-stage capacitive deionization (CDI) absorption air-conditioning system [14,15]. Reprinted with permission from ref. [14]. Copyright 2016 Li X., et al.

The energy demand of deionization is [14]:

$$P_{ad} = \frac{zF\Delta m_s}{\lambda M_s}U.  \tag{8}$$

where $U$ is the average voltage applied on CDI unit, $\lambda$ is the charge efficiency, $M_s$ is the molecular weight of the solute in the absorbent solution, $\Delta m_s$ is the mass flow rate of the solute ions adsorbed by the electrodes. In the regeneration process, the regenerator releases ions to the solution with no energy demand. On the contrary, it discharges the energy

stored when it works as the deionizer. Assume the energy recovery efficiency is $\eta$, and the energy consumption of the single-stage system is:

$$P_{1-CDI} = P_{ad} - \eta P_{ad} = (1 - \eta)\frac{zF\Delta m_s}{\lambda M_s}U. \tag{9}$$

The COP is given by [14,15]:

$$COP_{1-CDI} = \frac{l_w(1 - c_{or})\lambda M_s}{zFU(1 - \eta)c_{or}}. \tag{10}$$

Equation (10) presents the performance of the single-stage CDI regeneration air-conditioning system.

### 2.3. Double-Stage CDI Absorption Air-Conditioning System

A CDI device usually consists of many CDI units of different stages, which means the single-stage CDI system is only an ideal example with too much simplification. To analyze the practical situation, we need to investigate the performance of the multi-stage system and find out the influence of different parameters. Obviously, the double-stage system is the most typical one and could be the indicator for the multi-stage system analysis. The sketch of the double-stage system is shown in Figure 3. The main system is generally the same as that of the single-stage system, except the regenerator and deionizer are composed of two units, respectively. The diagram indicating the mass balance is depicted in Figure 4. The initial solute concentration out of absorber is $c_0$, and the mass flow rate is $m_0$; $c_1$, $m_1$ stand for the concentration and mass flow rate of the solution at the exit of unit 1, respectively; $c_2$, $m_2$ stand for the concentration and mass flow rate of the solution at the exit of unit 2, respectively; $\Delta m_{s1}$ and $\Delta m_{s1}$ are the solute mass flow rate released by unit 1 and 2, respectively.

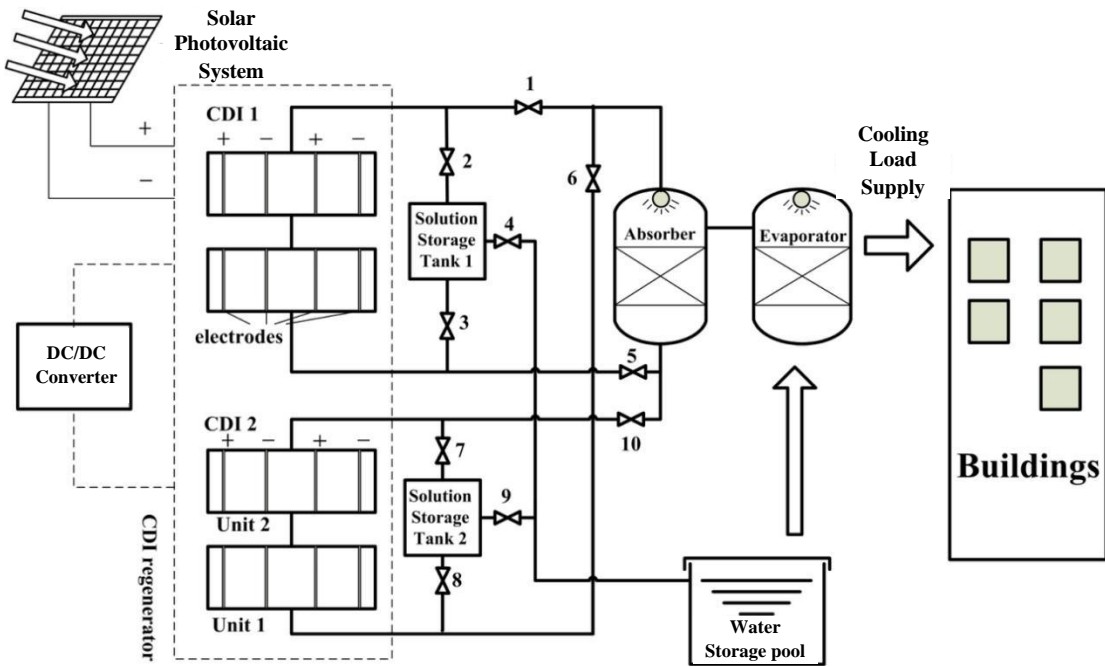

**Figure 3.** Flow chart of the double-stage CDI absorption air-conditioning system.

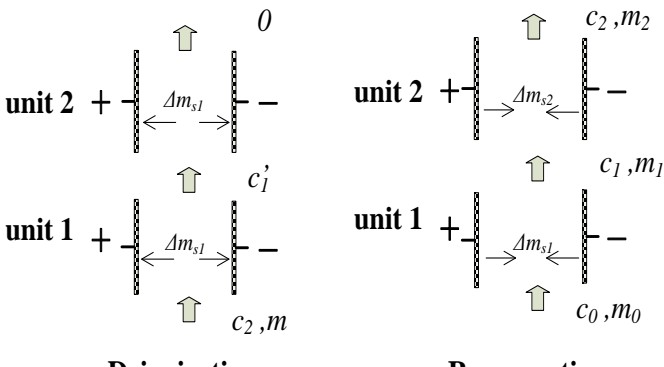

**Figure 4.** Mass relationship of double-stage system.

The energy consumption of the double-stage system is the sum of two CDI units:

$$P_{2-CDI} = (1-\eta)(P_{ad1} + P_{ad2}) = \frac{(1-\eta)zF}{\lambda M_s}(\Delta m_{s1}U_1 + \Delta m_{s2}U_2) \tag{11}$$

The mass balance of the regeneration process is:

$$m_0 + \Delta m_{s1} = m_1 c_1, \tag{12}$$

$$m_0 + \Delta m_{s1} = m_1, \tag{13}$$

$$m_1 c_1 + \Delta m_{s2} = m_2 c_2, \tag{14}$$

$$m_1 + \Delta m_{s2} = m_2. \tag{15}$$

The total solute mass flow rate released to the solution out of absorber is $\Delta m_s$:

$$\Delta m_s = \Delta m_{s1} + \Delta m_{s2} = \frac{c_2 - c_0}{1 - c_2}m_0. \tag{16}$$

In the absorption process, a strong absorbent solution absorbs water vapor to make the water in the evaporator keep on evaporating and affording a cooling load. After that, the concentration of absorbent solution decreases and it turns to a weak solution. There are the mass equations:

$$m_{ia}c_{ia} = m_{oa}c_{oa}, \tag{17}$$

$$m_{ia} + \Delta m_w = m_{oa}. \tag{18}$$

Combine Equations (17) and (18):

$$\Delta m_w = \frac{c_{or} - c_{oa}}{c_{or}}m_{oa} \tag{19}$$

where $\Delta m_w$ stands for the mass of water vapor absorbed per second; $m_{ia}$ is the mass flow rate of the solution at the entrance of the absorber and its concentration is $c_{ia}$; $m_{oa}$ stands for the mass flow rate of the solution at the exit of the absorber and its concentration is $c_{oa}$. The acquired cooling capacity $Q$ can be treated as the absorbed heat amount by evaporating $\Delta m_w$ kg water:

$$Q = l\Delta m_w. \tag{20}$$

The coefficient of performance of the system is:

$$COP_{2-CDI} = \frac{Q}{P_{2-CDI}} = \frac{\lambda M_s l_w \Delta m_w}{(1-\eta)zF(\Delta m_{s1}U_1 + \Delta m_{s2}U_2)}. \tag{21}$$

Assume $\Delta m_{s1} = k_1 \Delta m_s$, $\Delta m_{s2} = (1 - k_1) \Delta m_s$, combining Equations (16), (19) and (21), we can get the performance of the double-stage system:

$$COP_{2-CDI} = \frac{\lambda M_s l_w (1 - c_2)}{(1 - \eta) z F U_1 c_2} \frac{1}{(k_1 + (1 - k_1) U_2 / U_1)}. \tag{22}$$

The diagram indicating the mass balance of the multi-stage CDI is depicted in Figure 5. $c_i$ is the solute concentration at the exit of concentrated cells of unit $i$, $U_i$ is the voltage applied on unit $i$, $k_i$ is the mass transferred ratio between unit $i$ and 1. The COP of the multi-stage CDI system can be given in exactly the same way:

$$COP_{n-CDI} = \frac{\lambda M_s l_w (1 - c_n)}{(1 - \eta) z F U_1 c_n \sum\limits_{i=1}^{n} \frac{U_i}{U_1} k_i} \tag{23}$$

Base on Equations (22) and (23), analysis can be made on the double-stage or multi-stage system.

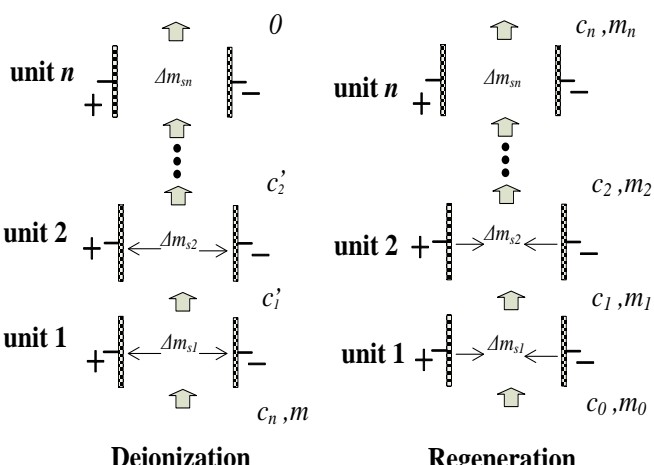

**Figure 5.** Mass relationship of multi-stage system.

## 3. Results and Discussion

### 3.1. The Adsorption Properties of CDI Method

Before exposing the performance of the double or multi-stage system, analysis has been made on the adsorption properties, which will be useful for the later investigation. The adsorption capacity can be obtained with Equation (7). It is easy to find the results are determined by three parameters: the potential difference between an electrode pair ($\varphi_0$), the temperature ($T$) and the initial concentration ($c_0$) of the electrolyte solution. Provided the concentration of the absorbent solution is 8 mol/L and the temperature is 298 K [14,15], Figure 6 presents the variation of the molar mass of the adsorbed ions in the unit area with different potential differences. The amount of adsorbed ions increases as the potential difference increases. For each 0.1 V increase in the potential difference, the amount of adsorbed ions is almost increased by an order of magnitude. That indicates increasing the potential greatly improves the adsorption effect, which is helpful to reduce electrode pairs. As the porous electrode is expensive, for the purpose of reducing the initial cost, the potential difference should be as high as possible below the potential causing electrochemical reactions.

Figure 7 shows the influence of the temperature. The values are calculated at different temperatures. Clearly, the adsorbed ions decrease as temperature rises. There are two reasons: First, the potential of the electrode decreases with the temperature rising, thus reducing the electric field force. So the ions adsorption capacity of the electrode surface is reduced. Second, the increase of the temperature strengthens ionic activity, its Brown movement is more intense and it is difficult to adsorb them by the electrode surface.

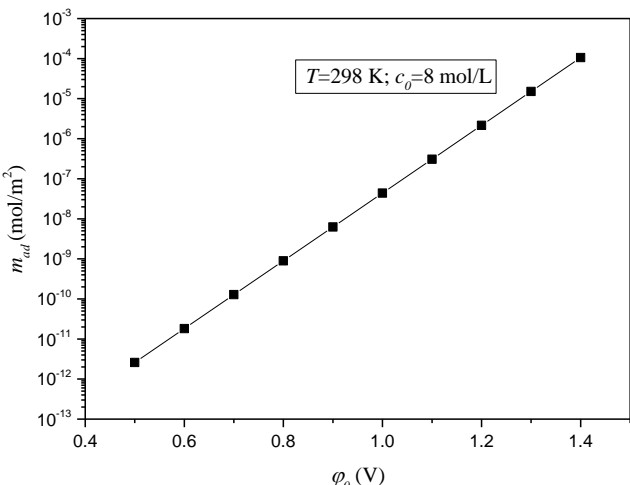

**Figure 6.** Adsorption capacity variation with different potentials.

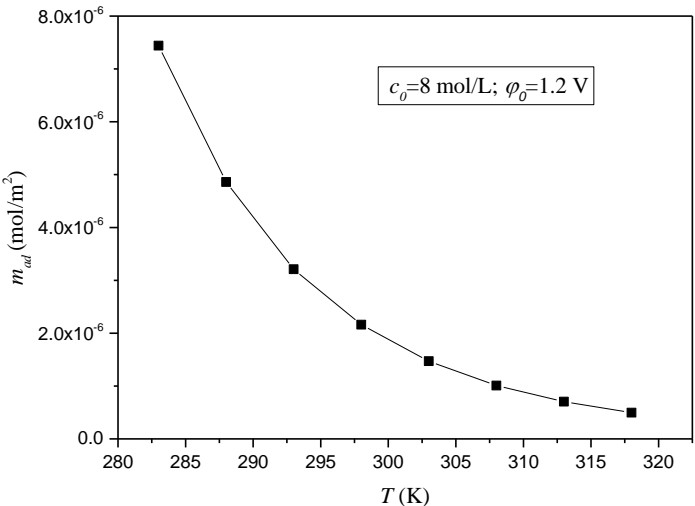

**Figure 7.** Adsorption capacity variation with different temperatures.

The effect of the initial concentration on the amount of adsorption is also great. It is calculated with five different concentrations. The results are displayed in Figure 8. With the concentration rising, more ions are adsorbed. As the solution concentration increases, the initial concentration gradient between the solution and the electrode becomes larger. So the driving force to push the ions toward the electrodes is greater, and the ions are more easily absorbed by the electrodes. However, as Equation (7) shows, the amount of the adsorbed ions is in proportion to the square root of the concentration. Therefore, the increase of the adsorption amount gradually decreases with the increase of the concentration.

From those figures, especially Figure 6, we can find the adsorption capacity is limited by the driving force or the applied voltage potential. However, this potential could not be too big or the chemical reaction happens. Thus, it may cause very long electrodes, which offers enough time for a complete adsorption process. It is definitely not a good idea as the length of the electrode could be in close relationship with its strength. To this aspect, a multi-stage system could be a necessary measure to overcome this difficulty. Moreover, different stages may use different voltages or other parameters, which may bring some changes or optimize the performance. Equation (7) is helpful in this work.

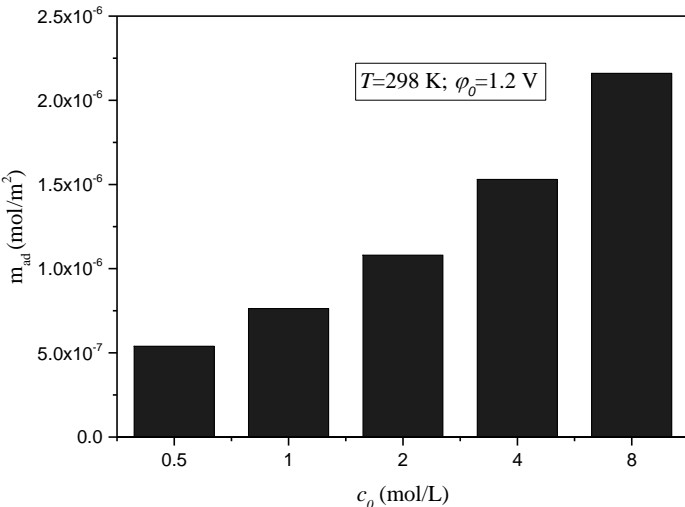

**Figure 8.** Adsorption capacity variation with different concentrations.

### 3.2. Performance Analysis of the Double-Stage CDI System

With Equation (22), COP of the double-stage CDI system can be calculated. Compared with the single-stage CDI system, COP of the double-stage CDI system is additionally impacted by two parameters, $k_1$ and $U_2/U_1$. The single-stage system could be taken as a special case of the double-stage system, if $k_1 = 1$ or $U_2/U_1 = 1$, there would be no differences between the two systems. For other parameters (which have been researched in related articles [14,15]), there are no essential differences between those of the single-stage system. Therefore, the following analysis mainly focuses on two parameters: $k_1$ and $U_2/U_1$.

Figure 9 shows the results. The used parameters are listed in Table 1. The range of $k_1$ is from 0.1 to 1, and the range of $U_2/U_1$ is from 0.4 to 1.6. The dark yellow surface stands for the COP of the single-stage system, while the colorful surface stands for the double-stage system. It can be seen the COP of the double-stage system varies greatly in the ranges, the maximum and minimum values are 4.8 and 2.1, respectively. On the contrary, the COP of a single-stage system is always 2.3.

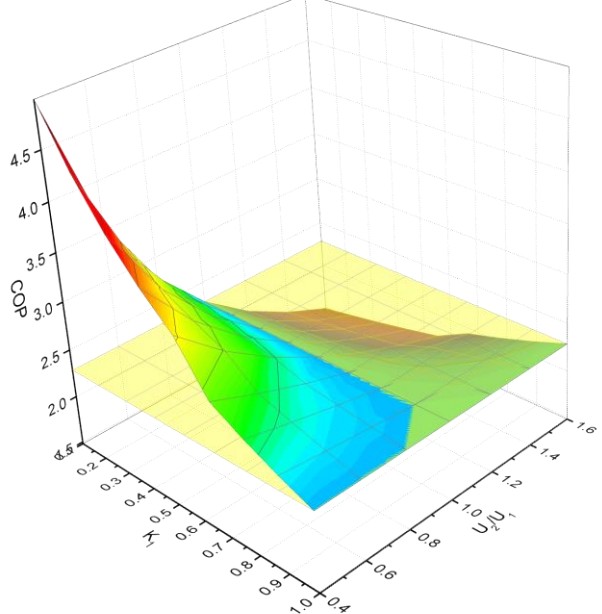

**Figure 9.** Coefficient of performance (COP) of the double-stage CDI system.

**Table 1.** Parameters for COP evaluation.

| Parameter | Value | Remarks |
|-----------|-------|---------|
| $c_2$ | 55% | |
| z | 1 | The absorbent is LiBr solution |
| $\lambda$ | 50% | This value is from a typical CDI product |
| $\eta$ | 50% | This value is from a typical CDI product |
| $U_1$ | 0.8 V | The common voltage of a CDI product |

It should be noted that $k_1$ and $U_2/U_1$ are not independent of each other by analyzing Equation (7): if the initial concentrations are totally the same, the higher potential applied on the electrodes, the more ions are adsorbed; if the potentials are equal, with lower initial concentration, fewer ions are adsorbed. We can infer as follows:

1. If $k_1 < 0.5$, it indicates the ions adsorbed by unit 1 are less than that of unit 2; on the other side, $c_1 < c_0$, so $U_2$ must be larger than $U_1$, it is $U_2/U_1 > 1$.
2. It can be found from Figure 9, the COP of the double-stage system is always less than that of the single-stage system when $U_2/U_1 > 1$.

Consequently, to make the double-stage system more efficient, $U_2/U_1$ must be less than 1. Then, what relation do the two parameters meet to make $U_2/U_1 < 1$? Firstly, we need to know the value of $k_1$ when $U_2/U_1 = 1$. If the solution temperature and the potential are equal ($U_2/U_1 = 1$), assume there are $m$ kg solution is diluted to be purified water, the concentration is $c_2$, the adsorbed solute is:

$$\Delta m_s = mc_2 \tag{24}$$

The amount of solute adsorbed by CDI unit 1 is:

$$\Delta m_{s1} = (k_1 mc_2) \tag{25}$$

The amount of solute adsorbed by CDI unit 2 is:

$$\Delta m_{s2} = (1 - k_1)mc_2 \tag{26}$$

Assume the concentration of the solution at the exit of unit 1 is $c_1'$, from the point of view of the mass balance of the solution, there is:

$$\Delta m_{s2} = (m - k_1 mc_2)c_1' \tag{27}$$

The concentration of the solution at the exit of unit 1 is:

$$c_1' = \frac{c_2 - k_1 c_2}{1 - k_1 c_2} \tag{28}$$

According to Equation (7), the adsorbed solute is:

$$\Delta m_{s1} = c_{ad} \cdot c_2^{1/2} \tag{29}$$

$$\Delta m_{s2} = c_{ad} \cdot c_1'^{1/2} \tag{30}$$

Combine Equations (24)–(30):

$$\frac{\Delta m_{s2}}{\Delta m_{s1}} = \left(\frac{c_1'}{c_2}\right) = \frac{1 - k_1}{k_1} = \left(\frac{1 - k_1}{1 - k_1 c_2}\right)^{1/2} \tag{31}$$

There is the solution of Equation (31):

$$k_1 = \frac{(1+c_2) - \sqrt{(1+c_2)^2 - 4(c_2-1)}}{2(c_2-1)} \tag{32}$$

The solution is calculated with $U_2/U_1 = 1$, and $0 < c_2 < 1$, so there is:

$$0.5 < \frac{(1+c_2) - \sqrt{(1+c_2)^2 - 4(c_2-1)}}{2(c_2-1)} < 0.61 \tag{33}$$

Consequently, in order to make $U_2/U_1 < 1$:

$$\frac{(1+c_2) - \sqrt{(1+c_2)^2 - 4(c_2-1)}}{2(c_2-1)} < k_1 < 1 \tag{34}$$

Provided the absorbent solution is LiBr, its typical working concentration range is 50–60%, correspondingly, there is $0.6 < k_1 < 1$. Then, what is the value of $U_2/U_1$? As Figure 6 shows, the adsorbed ions are almost increased by an order of magnitude as the potential increased by 0.1 V. Therefore, the difference between $U_2$ and $U_1$ is small unless $k_1$ is very close to 1. However, $k_1$ could not get close to 1 for practical using, so $U_2/U_1$ should be at least 0.8.

Figure 10 is obtained with the correction value. Compared with the single-stage system ($k_1 = 1$ or $U_2/U_1 = 1$), the double-stage system could be more efficient within certain ranges. However, the promotion is small, as the COP is only increased by 0.2 under the best conditions. However, still, more stages are beneficial. The multi-stage CDI unit can improve the system performance as long as the appropriate operating parameters are adopted. In other words, since the practical CDI device usually consists of many units, it is bound to have higher efficiency than the assumed single-stage model.

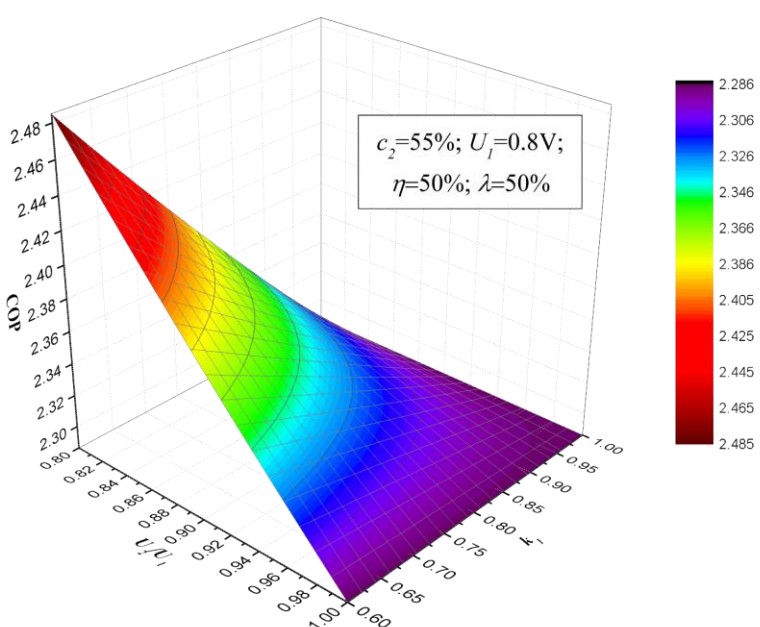

**Figure 10.** COP of the double-stage CDI system with correction parameters.

### 3.3. Optimization of the Multi-Stage CDI System

As for the multi-stage system, the affecting parameters are more complicated as shown in Equation (23). However, the analysis can be made by analogy with the double-stage system. It can be figured out two principles to follow for better performance. First, $k_1 > 1/n$,

otherwise, there must be a voltage of one unit higher than that of unit 1, which is not a good choice according to the analysis of the double-stage system. Second, $k_n < k_{n-1} < ... < k_2 < k_1$, in other words, the upstream unit needs to adsorb more ions than the next downstream unit. Like the double-stage system, the mass balance of each unit can be expressed, thus the adsorption ratio of each unit is obtained under the condition of the equal voltage applied on all the units. With the optimized parameters, the multistage system must be more efficient than the double-stage system. However, more stages will increase the cost and the complexity of the system. Whether it is worthwhile and how many stages is the best must be considered according to the practical situation.

## 4. Conclusions

CDI is an ion removal technique with high efficiency. It has the potential to replace the thermal regeneration method for the absorption air conditioning system. The CDI-based absorption system has a higher COP than the thermal-driven absorption system. It also has smaller size as CDI units replace the generator and save the auxiliary equipment like cooling tower. The electric driven chiller can be driven by a small-scale electric generator, it has higher efficiency, along with better characteristics and load followability. To reveal the influences caused by multiple CDI units, the adsorption properties of electrodes are simulated and the double-stage system is established as a typical model of the multi-stage system.

It is found that the potential, the temperature of the solution and the initial concentration have great impacts on the adsorption capacity of the electrodes. The potential is the key factor. Through the analysis of the double-stage system, it exposes that the parameters should be limited to a certain range to make the system more efficient. When the absorbent is LiBr solution, the adsorption ratio of unit 1 should be in the range of 0.6–1, correspondingly, the voltage ratio is in the range of 0.8–1. Although the improvement is not much (the COP is only increased by 0.2 as the simulation shows), the more stages system has great potential to improve the performance with proper parameters. As an analogy with the double-stage system, the optimization parameters for the multi-stage system have been ascertained. The performance will increase with more stages, while it also calls for most cost. A better plan is to consider both the performance improvement and the cost increase according to the situation. More experimental work will be performed in the future to expose the actual performance of the double or multi-stage system.

**Author Contributions:** Conceptualization, X.L.; methodology, F.C.; validation, B.D.; formal analysis, F.C.; investigation, F.C.; resources, X.L.; writing—original draft preparation, F.C.; writing—review and editing, X.L.; supervision, X.L.; project administration, X.L.; funding acquisition, X.L. All authors have read and agreed to the published version of the manuscript.

**Funding:** This research was funded by the fund of National Natural Science Foundation of China under the contracts No. 51206080 and No. 51676098; the fund of Natural Science Foundation of Jiangsu Province under the contracts No. BK20170095 and No. BK20160822.

**Institutional Review Board Statement:** Not applicable.

**Informed Consent Statement:** Not applicable.

**Data Availability Statement:** Data is contained within the article.

**Acknowledgments:** This research was supported by the grants from the fund of National Natural Science Foundation of China under the contracts No. 51206080 and No. 51676098. It was also supported by the fund of Natural Science Foundation of Jiangsu Province under the contracts No. BK20170095 and No. BK20160822. These supports are gratefully acknowledged.

**Conflicts of Interest:** The authors declare no conflict of interest.

## Nomenclature

| | |
|---|---|
| c | Mass concentration of the solute |
| F | Faraday constant, sA/mol |
| *lw* | Latent heat of evaporation of water, kJ/(kg × K) |
| *Ms* | Molecular weight of the solute in the absorbent solution, kg/mol |
| U | Voltage, V |
| z | Electrochemical valence |
| P | Energy consumption |
| Q | Acquired cooling capacity, kJ |
| m | Mass flow rate, kg/s |
| T | Temperature, K |

**Greek letters**

| | |
|---|---|
| φ | Potential, V |
| ρ | Charge density, $C/m^2$ |
| λ | Charge efficiency |
| η | Energy recovery efficiency |

**Subscripts**

| | |
|---|---|
| 0 | Initial solution |
| 1 | Solution at exit of CDI unit 1 |
| 2 | Solution at exit of CDI unit 2 |
| ia | Entrance of absorber |
| oa | Exit of absorber |
| n | Number of stages |

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
