# Peer review of "Research on Double-Stage and Multi-Stage Capacitive Deionization Absorption Air-Conditioning System"

_processes, doi:10.3390/pr9020395_

Round 1

Reviewer 1 Report

In this manuscript by Feng Cheng et al., the authors present research on double-stage and multi-stage capacitive deionization absorption air-conditioning system. In my opinion the reviewed paper definitely qualifies for publication in Processes journal. However, the authors should address the following issues before this work can be accepted for publication.

  1. Page 1, line 23. Performance ---> double-stage and multi-stage systems
  2. The all manuscript. Dots after “]” – see for example page1, lines 38, 41, 44.
  3. Page 1, line 44. COP?
  4. Page 2, line 69, “The present work only…”. New paragraph.
  5. Page 3, Fig. 1. (Capital letter) load ---> Load.
  6. Page 3, Eqs. (1)-(7). Reference(s)?
  7. Page 4, Fig. 2. (Capital letter) Solar photovoltaic system ---> Solar Photovoltaic System. Cooling load supply ---> Cooling Load Supply.
  8. Page 4, line 107. [14,15] - no superscript.
  9. The all manuscript. For example, Pages 5 (lines 150 – 154) 8 (line 176). Problem with subscripts – c0, m0….
  10. Page 6, Fig. 3. (Capital letter) Solar photovoltaic system ---> Solar Photovoltaic System. Cooling load supply ---> Cooling Load Supply. DC/DC converter ---> DC/DC Converter. Water storage pool ---> Water Storage Pool.
  11. 1 and 4-5. Standardize the order of showing “Regeneration” and “Deionization”, i.e. in Fig. 1 (1st “Deionization” and 2nd “Regeneration”) Figs. 4 and 5 (1st “Regeneration” and 2nd “Deionization”).
  12. Page 7, line 160. double ---> multi.
  13. The all manuscript. For example, Pages 7 (lines 170-174) 8 (lines 185-187). Badly formatted text.
  14. Page 14, Subscripts. Please add “n”.

Author Response

All the revisions related to Reviewer #1s comments have been marked with cyan color in my article: e.g. L.

Reviewer #1: In this manuscript by Feng Cheng et al., the authors present research on double-stage and multi-stage capacitive deionization absorption air-conditioning system. In my opinion the reviewed paper definitely qualifies for publication in Processes journal. However, the authors should address the following issues before this work can be accepted for publication.

  1. Page 1, line 23. Performance ---> double-stage and multi-stage systems

Revision: It has been revised in the new manuscript.

  1. The all manuscript. Dots after “]” – see for example page1, lines 38, 41, 44.

Revision: It has been revised in the new manuscript.

  1. Page 1, line 44. COP?

Revision: It has been revised in the new manuscript.

  1. Page 2, line 69, “The present work only…”. New paragraph.

Revision: It has been revised in the new manuscript.

  1. Page 3, Fig. 1. (Capital letter) load ---> Load.

Revision: It has been revised in the new manuscript.

  1. Page 3, Eqs. (1)-(7). Reference(s)?

Revision: It has been revised in the new manuscript.

  1. Page 4, Fig. 2. (Capital letter) Solar photovoltaic system ---> Solar Photovoltaic System. Cooling load supply ---> Cooling Load Supply.

Revision: It has been revised in the new manuscript.

  1. Page 4, line 107. [14,15] - no superscript.

Revision: It has been revised in the new manuscript.

  1. The all manuscript. For example, Pages 5 (lines 150 – 154) 8 (line 176). Problem with subscripts – c0, m0….

Revision: It has been revised in the new manuscript.

10.Page 6, Fig. 3. (Capital letter) Solar photovoltaic system ---> Solar Photovoltaic System. Cooling load supply ---> Cooling Load Supply. DC/DC converter ---> DC/DC Converter. Water storage pool ---> Water Storage Pool.

Revision: It has been revised in the new manuscript.

11.1 and 4-5. Standardize the order of showing “Regeneration” and “Deionization”, i.e. in Fig. 1 (1st “Deionization” and 2nd “Regeneration”) Figs. 4 and 5 (1st “Regeneration” and 2nd “Deionization”).

Revision: It has been revised in the new manuscript.

  1. Page 7, line 160. double ---> multi.

Revision: It has been revised in the new manuscript.

  1. The all manuscript. For example, Pages 7 (lines 170-174) 8 (lines 185-187). Badly formatted text.

Revision: It has been revised in the new manuscript.

14.Page 14, Subscripts. Please add “n”.

Revision: It has been revised in the new manuscript.

Thank you!

Reviewer 2 Report

Your study is a very important. Therefore, I think it is important that this research is reflected in practice. So, I think it is necessary to show the following points.

Shouldn't it be suggested as the next step, at least in the conclusion?

1. About performance

  1. Relationship between temperature of chilled water obtained from the evaporator and the COP
  2. Relationship between part-load ratio and COP
  3. Advantages over thermal-driven absorption chillers at COP, including auxiliary equipment inside the equipment and auxiliary equipment outside the equipment (for example, cooling towers)
  4. Superiority over electric driven chiller (COP, partial load characteristics, load followability, etc.)

2. Information as a product

      Equipment size compared to thermal-driven absorption chillers

3. System

      The solar power supply is drawn as an input in Figures 2 and 3.

      Is this required?

That's all, thank you very much.

Author Response

All the revisions related to Reviewer #2s comments have been marked with magenta color in my article: e.g. L.

Your study is a very important. Therefore, I think it is important that this research is reflected in practice. So, I think it is necessary to show the following points.

Shouldn't it be suggested as the next step, at least in the conclusion?

  1. About performance

a.Relationship between temperature of chilled water obtained from the evaporator and the COP

Revision: With the temperature decreasing of the produced chilled water, cooling load will increase. That means more water vapor (refrigerant) would be absorbed by the absorbent solution, which further reduces the concentration of the solution to be concentrated. This will improve the efficiency as the COP is higher with lower concentration of the regenerated solution. It has been added in the introduction part of the new manuscript.

b.Relationship between part-load ratio and COP

Revision: With the increasing of part-load ratio, more water vapor (refrigerant) would be absorbed by the absorbent solution, which reduces the concentration of the solution to be concentrated. This will improve the efficiency as the COP is higher with lower concentration of the regenerated solution. It has been added in the introduction part of the new manuscript.

c.Advantages over thermal-driven absorption chillers at COP, including auxiliary equipment inside the equipment and auxiliary equipment outside the equipment (for example, cooling towers)

Revision: The new system has higher COP than the thermal-driven absorption system. CDI units replace the generator and save the auxiliary equipment like the cooling tower It has been added in the conclusion of the new manuscript.

d.Superiority over electric driven chiller (COP, partial load characteristics, load followability, etc.)

Revision: This electric driven chiller has higher COP, along with better characteristic and load followability. It has been added in the conclusion of the new manuscript.

  1. Information as a product

Equipment size compared to thermal-driven absorption chillers

Revision: Without generator and auxiliary equipment, the equipment size of the new system is smaller than thermal-driven absorption system. It has been added in the conclusion of the new manuscript.

  1. System

The solar power supply is drawn as an input in Figures 2 and 3. Is this required?

Revision: The new system can be driven by small scale electric generator, the solar power is one example. It has been added in the conclusion of the new manuscript.

That's all, thank you very much.

Thank you!
